# Delay in Diagnosis and Treatment of Primary Bone Tumors during COVID-19 Pandemic in Poland

**DOI:** 10.3390/cancers14246037

**Published:** 2022-12-08

**Authors:** Daniel Kotrych, Dawid Ciechanowicz, Jakub Pawlik, Martyna Szyjkowska, Bartłomiej Kwapisz, Maciej Mądry

**Affiliations:** 1Department of Children Orthopaedics and Musculoskeletal Oncology, Pomeranian Medical University, 71-281 Szczecin, Poland; 2Department of Orthopaedics, Traumatology and Musculoskeletal Oncology, Pomeranian Medical University, 71-281 Szczecin, Poland; 3Department of Orthopaedics, Traumatology and Orthopaedic Oncology, Collegium Medicum University in Zielona Gora, 65-046 Zielona Gora, Poland

**Keywords:** diagnosis delay, bone sarcoma, bone tumor, COVID-19, symptoms, biopsy

## Abstract

**Simple Summary:**

The diagnosis and treatment of patients with bone neoplasms is often a demanding task, and at the same time, inadequate treatment may worsen the patients’ prognosis. During the COVID-19 pandemic, the change of priorities caused a limitation or even a temporal pause of elective healthcare in medical facilities, which included: delays in or rescheduling of outpatient visits, planned diagnostic and therapeutic procedures, as well as a shift towards telemedicine, which precluded the proper physical examination of the patients. Therefore, the aim of our study was to determine how the pandemic influenced the diagnostic and therapeutic processes for patients with primary bone tumors, and what lessons we can learn for the future. In addition, our goal was to verify whether surgical treatment during the COVID-19 pandemic could be safe for patients with bone tumors. In our study, we did not observe an extension of the delay in diagnosis during the COVID-19 pandemic, however, those patients reported to oncology centers with larger tumors compared to those of the patients who were treated before the pandemic. The COVID-19 pandemic had a negative impact on the delay in performing biopsies, however, it did not significantly affect the therapeutic process itself. At the same time, we indicated that the patients with bone tumors can be successfully treated surgically without experiencing an unnecessary delay, if all the precautions are taken.

**Abstract:**

Background: The COVID-19 pandemic has affected all of the medical specialties, including orthopedic oncology. Therefore, the aim of the study was to assess how it influenced the diagnostic and therapeutic processes for patients with bone neoplasms. Methods: We evaluated 87 patients treated due for bone neoplasms before (Group I, *n* = 36) and during the COVID-19 pandemic (Group II, *n* = 51). A delay in diagnosis was defined as the period between the initial clinical symptoms and the date of referral to an oncology center. The patients from Group II were asked to complete a short questionnaire regarding the COVID-19 pandemic. Results: The median general delay in diagnosis before the pandemic was 7 months, while during the pandemic, it was 10 months (*p* = 0.728). The biopsy delay was lower in the pre-pandemic group: median-6.5 vs. 12 days (*p* = 0.025). The patients from Group II were diagnosed with larger tumors compared to those in Group I: the median values were 75 vs. 56 mm (*p* = 0.025), respectively. After an X-ray examination, the bone neoplasms were suspected more frequently in the Group II: 63% vs. 44% cases (*p* = 0.024), respectively. In Group II, 20 (60.8%) cases of SARS-CoV-2 infection were reported, however, no respiratory failure cases were noticed. Conclusion: The pandemic affected the diagnostic process of primary bone tumors, resulting in delays in performing biopsies. During the pandemic, the patients reported larger diameters of their bone lesions.

## 1. Introduction

The COVID-19 pandemic is an ongoing, global outburst of the novel coronavirus (SARS-CoV-2). It was officially acknowledged by the World Health Organization on 11 March 2020. Since then, over 565 million cases of SARS-CoV-2 infections have been reported, and 6,373,739 people have died [1]. Yet, all of these casualties are a direct effect of SARS-CoV-2’s influence. There is also an indirect effect—the number and rate of new COVID-19 cases was so great at times that new priorities in healthcare must have been implemented in order to manage the pandemic. As they helped to resolve to COVID-19 situation, they had a detrimental effect on other aspects of people’s health. The shift in priorities caused a limitation or even a temporal pause of elective healthcare in medical facilities, which included: delays in or the rescheduling of outpatient visits, planned diagnostic and therapeutic procedures, as well as a shift towards telemedicine, which precludes the proper physical examination of the patient [1,2]. All of the above might be a cause of doctor-related delay in diagnosis and treatment [2,3,4]. On the other hand, the patients fearing potential exposure to SARS-CoV-2 might have decided not to visit their doctor or refused to register for planned procedures or imaging tests (X-ray, MRI, etc.), thus causing a patient-related delay [3,5].

For oncology patients, it is known that delay in obtaining the diagnosis results in a higher clinical stage and a poorer prognosis after the diagnosis [6,7]. Moreover, it has a direct impact on the treatment response and survival of the patients [8,9]. We have already seen reports of the COVID-19 pandemic on extended delays in diagnoses and a reduction of the number of newly diagnosed cases of head and neck, gastrointestinal, colorectal cancer as well as bone tumors [10,11,12,13]. The pandemic has caused a delay in not only diagnosis, but also the treatment of oncological patients, finally resulting in an increased number of potentially avoidable cancer deaths [14,15]. Additionally, it is expected that as seen in other times of crisis (i.e., natural/economical disasters or disease outbursts), we will see a surge in the number of new cases of malignancies, most likely at a more advanced stage [16].

Delay in the diagnosis of bone tumors are a current problem. A small number of patients and non-specific symptoms already existed before the COVID-19 pandemic, and the studies reported a delay during this period of 6–10 months on average [17,18,19]. On the other hand, the studies about the impact of the COVID-19 pandemic on the diagnosis and treatment of patients with bone neoplasms showed an increase in the length of the treatment delay (in the beginning of chemotherapy, radiotherapy and surgical treatments) and a longer delay in performing biopsy [20,21]. However, so far, the treatment of patients with bone tumors during the pandemic has not been reported to negatively impact the treatment outcomes and general survival of these patients [21]. To the best of our knowledge, there are still lack of reports on how the COVID-19 pandemic has affected the pre-hospital diagnosis and treatment of patients with bone tumors. Therefore, our goal in this study is to determine if the pandemic has extended the delays in any way, and if so, how to prevent them in the future.

## 2. Materials and Methods

The patients treated in the Department of Orthopedics, Traumatology and Musculoskeletal Oncology for neoplastic bone lesions in the period between January 2018 and March 2022 were invited to the study. All of the patients were divided into two groups; the patients treated and diagnosed before the COVID-19 pandemic (until 10 March 2020)—Group I; the Patients treated and diagnosed during the COVID-19 pandemic (from 11 March 2020)—Group II. The exclusion criteria were as follows: (1) the diagnosis of a secondary bone neoplasm (bone metastases); (2) patients with primary bone neoplasms showing no symptoms, which were detected by chance; (3) the recurrence of the neoplastic lesion in bone after a previous treatment. The minimum follow-up length was 6 months. All of the patients or their relatives gave written informed consent to be included in scientific study during their admission to the hospital. All of the procedures performed in studies involving human participants were in accordance with the Ethical Standards of the Institutional and National Research Committee and with the 1964 Helsinki Declaration and its later amendments or comparable ethical standards, and it has been approved by the authors’ institutional review board.

Based on the patients’ medical history, we collected data about the delays in diagnoses: the symptoms that were reported at the first presentation to the physician, the overall duration of the symptoms (the time from the onset of symptoms to the first presentation to the physician—the ‘patient-related delay’). Next, we evaluated the duration of the symptoms from time of the first visit to the GP (*General practitioner*) to the suspicion of a bone neoplasm and referral to an oncology center— the ‘doctor-related delay’. The term ‘general delay in diagnosis’ was used as to describe the time from the occurrence of the first symptoms to referral to a oncology center. From the analysis of the medical records, we obtained information about what radiology examinations the patients had ordered before their referral to our department. Further, we checked what percentage of these examinations indicated the presence of a bone lesion. The diagnostic and therapeutic pathway in our department was also analyzed: the time from their referral to the orthopedic oncology center to the date of the performance of the biopsy—the ‘time to perform the biopsy’; the period from the biopsy to the result of the histopathological examination—‘time to histopathological result’; the time from receiving the histopathological result to the date of the surgery—‘time to perform surgery’. To define the characteristics of the groups, the following data were also collected: age at the time of the suspicion of a bone tumor, gender, the primary localization of the lesion, the histopathological type of the bone neoplasms and also the largest size of the tumor at the time of their referral to our center. Moreover, we checked in what percentage limb-sparing surgeries were performed on the patients.

All of the patients treated during the COVID-19 pandemic were asked to answer the questions in the anonymous Author’s COVID-19 Specific Questionnaire [Appendix A]. The survey was created based on the authors’ clinical experience and previous publications about the impact of the COVID-19 pandemic on the delay in the diagnosis and treatment of patients with bone tumors [20,22]. The validity of the survey was not tested before the study. It contained 9 closed-ended questions (yes or no questions), and its aim was to determine, according to the patients, how the pandemic affected their diagnosis and treatment process. First, the patients were asked if they had been diagnosed with the COVID-19 infection during the bone neoplasm’s diagnostic or treatment process (symptoms and/or a positive PCR test result). Another question referred whether the infection was complicated by respiratory failure. The patients were also asked if during the pandemic they had experienced difficulties with access to a general practitioner or orthopedist for radiological examinations. Subsequent questions concerned whether the biopsy, chemotherapy (treatment) or surgery had to be postponed due to the pandemic. The last question checked whether the fear of the COVID-19 disease influenced their later visits to the physician. The survey was completed during the follow-up visit, 6 months after the surgery or it was sent online to the patients. Each of the patients agreed to complete the questionnaire. Providing none of the answers or selecting two answers in one question disqualified the patients survey from the study.

The data were collated into a separate file in Microsoft Excel 2019 (Microsoft Corporation, Redmond, WA, USA). The statistics steps were performed using the Statistica 13.0.2 program (StatSoft Polska Co. Ltd., Kraków Poland). The Mann–Whitney U test was used to compare the collected data (the delay in diagnosis and the size of the lesion) between the groups (Group I vs. Group II) because the data distribution had the characteristics of an abnormal distribution (Shapiro–Wilk test). The Chi2 test was used to compare the data relating to: gender, the type of bone neoplasm, the localization of the tumor, the symptoms and the radiological examinations. The data were also analyzed separately for malignant bone neoplasms and benign bone neoplasms. The distribution was made on the basis of the 2020 WHO classification of bone and soft tissue neoplasms [23]. The level of significance was set at *p* < 0.05.

## 3. Results

### 3.1. Participant Characteristics

We evaluated 87 patients (M = 44; F = 43). To Group I, we added 36 patients (M = 17; F = 19), while to Group II, we added 51 patients (M = 27; F = 24). The mean age at the time of admission to the orthopedic oncology center in the study groups was: 40.7 (range: 16–87) years and 45.6 (range: 16–90) years, respectively. The most frequent type of neoplasms in both of the groups were: Chondrosarcoma with eleven (30.6%) vs. sixteen patients (31.4%), which was followed by Osteosarcoma with eight (22.2%) vs. thirteen patients (25.5%) and Giant Cell Tumor with six (16.7%) vs. six patients (11.8%), respectively. We classified the individual cases of malignant bone lesions (Ewing Sarcoma, Fibrosarcoma, Chordoma and Undifferentiated Pleomorphic Sarcoma) into the ‘Other malignant neoplasms’ category, while the benign bone lesions (Fibroma and Chondroma) were sorted into the ‘Other benign neoplasms’ category. The bone neoplasms most frequently affected the lower limb- in nineteen (52.8%) vs. twenty-five (49%) cases, which was followed by the upper limb in nine (25%) vs. thirteen (25.5%) cases, respectively, and the pelvis and the sacrum in eight (22.2%) vs. thirteen (25.5%) cases, respectively. During the pandemic, the patients who reported to the orthopedic oncology center had larger bone lesions compared to those in the pre-pandemic group: 56 vs. 75 mm, respectively (*p* = 0.025) (Table 1).

Analyzing the number of patients with bone neoplasms treated in our center, we observed a decrease in the number of cases during the COVID-19 pandemic. Before the pandemic, 164 patients with bone tumors were treated in our center, while from March 2020 to March 2022, there were about 94 patients (*p* = 0.000004). In 2018, the number of patients was 82, while in 2019, it was 69 (*p* = 0.0606). Subsequently, in 2020, we observed a decrease in the number of patients to 47 cases, while in 2021, the number of patients with bone tumors was 45 (*p* = 0.5067). In the first three months of 2022, 13 patients with bone neoplasms were treated in our center (Figure 1).

### 3.2. Delay in Diagnosis

There were no statistically significant differences in the delay in diagnosis in both of the groups. The median general delay in Group I was 7 months, while in Group II, it was 10 months. The differences between the groups were also not significant when we divided the patients into malignant and benign neoplasm cases. However, before the COVID-19 pandemic, the patients decided to visit a physician earlier compared to the patients during pandemic (the median values: 1 months vs. 2 months, respectively) (*p* = 0.619). On the other hand, the physicians took longer to diagnose the patients before the pandemic (the median values: 5 vs. 3 months, respectively) (*p* = 0.265) (Table 2). In both of the groups, the patients were referred to an orthopedic oncology center, most often within 4–12 months from the onset of the first symptoms (45.7% in Group I and 55.6% in Group II) (*p* = 0.869). Before the pandemic, the patients were more often diagnosed with bone tumor within 1 month from the onset of the first symptoms compared to the patients who were treated during the COVID-19 pandemic: 17.1% vs. 6.7% (*p* = 0.052), respectively. A similar number of patients were admitted to an orthopedic oncology center within 1–4 months: 20.0% vs. 22.2% (*p* = 0.483) in both of the groups, respectively. A delay of more than 12 months occurred in 17.1% of the patients before the pandemic, and in 15.5% of them during the pandemic (*p* = 0.636) (Figure 1).

### 3.3. Diagnosis and Treatment

The patients’ medical records were also analyzed in terms of the previously performed radiological examinations. In Group I, they were ordered in 24 (66.7%) cases during the first visit to the physician, while in Group II, there were 32 (39.5%) cases (*p* = 0.850). Before the pandemic, the most frequently ordered examinations were: X-ray (*n* = 18, 75%), MRI (*n* = 5, 20.8%) and CT (*n* = 1, 4.2%), whereas during the pandemic, the most frequently ordered examinations were: X-ray (*n* = 27, 84.4%), MRI (*n* = 4, 12.5%) and CT (*n* = 1, 3,15). The plain radiographs indicated bone neoplasms in eight (44%) cases in Group I and in 17 (63%) cases in Group II (*p* = 0.024). All of the other radiological examinations revealed bone neoplasms, and the patients were referred to an oncology center. The patients from Group I had a performed biopsy on the suspected bone lesion earlier compared to that in Group II: the median values were 6.5 vs. 12 days, respectively (*p* = 0.025). The histopathological results were also available earlier in Group I compared to in Group II: the median values were 12 vs. 13.5 days, respectively (*p* = 0.450). The surgical treatments were performed earlier before the pandemic compared to for the patients treated during the COVID-19 pandemic: 95 vs. 104 days, respectively (*p* = 0.494) (Table 2). Additionally, we noticed more amputations being performed in Group II compared to in Group I: seven (13.7%) vs. two (5.6%), respectively (*p* = 0.382).

### 3.4. Symptoms

The medical records also provided data about the symptoms which the patients reported during their first presentation to a physician. In both of the groups, the most common symptoms were: pain at the tumor site (*n* = 73), a swelling/palpable lump (*n* = 35), a disturbance in limb function (*n* = 20), a pathological fracture (*n* = 14), discomfort (*n* = 7) and general symptoms (*n* = 6) such as fever, weakness and weight loss (Figure 2). The statistical analysis also included the delay in the diagnosis between the patients reporting a specified symptom and not reporting this symptom. It is shown that the patients who presented for the first time to a physician/Emergency Department with a pathological fracture had a longer doctor-related delay compared to the group without a confirmed fracture: the median values were 8 vs. 3 months, respectively (*p* = 0.037). For the other symptoms, no differences in the length of the delay in the diagnosis were found between the groups (*p* > 0.05).

### 3.5. The COVID-19 Patient’s Survey

In the group of patients treated during the COVID-19 pandemic, 20 (39.2%) cases of SARS-CoV-2 infection were reported. However, a complicated course of infection (respiratory failure) was not reported in a single case. According to five (9.8%) patients, the pandemic significantly impeded their access to a general practitioner (GP), and also five (9.8%) patients reported a problem with gaining access to an orthopedist. Seven (13.7%) patients reported problems with gaining access to radiological examinations (X-ray, MRI, CT and USG) due to the pandemic. However, only one patient (2.0%) reported that if it were not for the COVID-19 pandemic, he would have visited the physician earlier. In the further diagnostic process, six (11.8%) patients reported a delay in having a biopsy due to the COVID-19 pandemic. However, in the treatment process, nine (17.6%) patients had a delay in the beginning of their treatment (neoadjuvant chemotherapy), and six (11.8%) patients reported a delay in performing the surgery (Figure 3).

## 4. Discussion

The delay in the diagnosis of primary bone neoplasms is a still significant problem. As our research shows, even before the COVID-19 pandemic, the median delay was 7 months. For comparison, George A. et al. reported a median delay of 5.9 months, and Ashwod et al. reported a delay of 7.5 months, while in the case of pelvic tumors, the delay was on average 10 months [17,24]. The early diagnosis of bone sarcoma gives a greater chance of performing limb-sparing surgery, and it gives a better functional and survival results. Poudel at el. assessed the risk factors associated with the local recurrence of bone sarcoma, and they found that each week of delay in the surgery increases the risk of recurrence by 1.14 times [25]. On the other hand, Goryń T. et al. indicated that the patients with the local recurrence of bone sarcoma have a significantly lower 5-year overall survival compared to the patients without recurrence: 30% vs. 70%, respectively [26]. As the current reports show, during the COVID-19 pandemic, the delay in cancer diagnosis and treatment has generally worsened [10,12,27]. Additionally, in our study, the median general delay in diagnosis during the pandemic increased from 7 months to 10 months. This trend was also visible in patient-related delays, where the median increased from 1 to 2 months. However, according to the data from survey from our study, only one patient reported that, if it were not for the COIVD-19 pandemic, he would have visited a physician earlier. On the other hand, about one in ten patients reported having restricted access to a GP and orthopedist, which could also have contributed to the prolongation of the patients-related delays. We also observed that more radiological examinations were ordered in the group of patients diagnosed before the pandemic compared to in the pandemic group (67% vs. 40%, respectively). This could be caused by their restricted access to these examinations, which was reported by 14% of the patients in the COVID-19 pandemic group. We reported in our study, a lower median of the doctor-related delay in group treated during the COVID-19 pandemic compared to the patients who were treated before the pandemic (3 months vs. 5 months, respectively). The diagnosis of bone neoplasms is often difficult because of the long period of the occurrence of nonspecific symptoms [17]. However, the more advanced the disease is, the clearer the diagnosis becomes. Hence, we suspect that in our study, when the patients decided to see a physician for the first time, they already presented a clear picture of the disease. This thesis may be confirmed by the fact that patients who were treated during the pandemic presented a larger diameter of the lesion (75 vs. 56 mm, respectively), and they had a palpable lump more frequently than the patients who were treated before the pandemic did (43% vs. 33%, respectively). It was also observed in our study that before the pandemic, more patients came to our center within 1 month after the first symptoms compared to the patients in the COVID-19 pandemic group (17% vs. 7%, respectively). According to Goedhart et al., the delay in the diagnosis of primary bone neoplasms should not exceed 4 months [18]. In our study, this rule was followed by approximately 37% of the patients before the pandemic and approximately 29% of the patients during the COVID-19 pandemic. All of the differences presented above are similar for both the malignant and benign bone neoplasms.

Another important issue that we verified in our study is the impact of the COVID-19 pandemic on the delay in performing biopsies and surgeries. Without a proper biopsy and histopathological result, the treatment should not be initiated in patients with suspected primary bone sarcoma. Moreover, the biopsy should be performed as soon as possible after their referral to an oncology center [28]. As the SONCOS guidelines suggest, the treatment should be started after a maximum of 42 days after the first oncological consultation [18]. We observed a significant longer delay in performing the biopsies in the group of patients treated during the COVID-19 pandemic (the median values: 12 days vs. 6.5 days, respectively; *p* = 0.025). We also indicated that the COVID-19 pandemic was the cause of the postponement of the biopsy in 12% of the cases. The time until obtaining a histopathological result was not observed to increase during the pandemic, and it was 12 days before the pandemic and 13.5 days during the pandemic, respectively. Additionally, the time to the surgery did not differ statistically in both of the groups (the median values: 95 vs. 104 days, respectively). However, the COVID-19 pandemic caused the postponement of surgery in about 12% of the cases. It is worth noting that in approximately 18% of all of the patients, the initiation of treatment (neoadjuvant chemotherapy and/or radiotherapy) was delayed due to the COVID-19 pandemic. As reported by Thaler M. et al. in a worldwide survey study, around 20% of surgeons canceled or postponed the surgery for patients with life-threatening sarcomas during the COVID-19 pandemic [22]. They also observed a delay or cancellation of the biopsy procedures in 20% of patients with musculoskeletal tumors, as well as a delay in the initiation of neoadjuvant chemotherapy (4.9% of the cases) and radiotherapy (11.7% of the cases).

The COVID-19 pandemic may have had a negative impact on the survival rate of oncological patients [29,30]. However, in our group, we did not observe increased mortality due to SARS-CoV-2 infections. Approximately 39% of the patients with bone sarcoma reported that they had symptoms and a positive PCR test for COVID-19 during the treatment process. However, no acute respiratory failure was reported, and all of the patients returned to the treatment after a quarantine period. Similar data are presented by Kumar VS. et al. who showed that the treatment of the patients with bone sarcomas during the COVID-19 pandemic can be safe for both the patients and medical staff, while maintaining all of the rules for personal protection [20]. Yet, it is worth remembering that the patients undergoing surgery have an increased risk of undergoing mechanical ventilation due to COVID-19 compared to the population of non-operated patients (20.6% vs. 3.6%, respectively) [29]. As reported by Stevenson et al., if all of the safety measures are implemented, it is possible to reduce the risk of a severe complications of SARS-CoV-2 infections in patients who are being treated surgically to a minimum [30]. On the other hand, Rajasekaran RB. et al. showed that in the case of the patients treated in a COVID-free facility, the number of the cases of infections and the number of complications associated with it were much lower than they were in standard hospital wards (1:3) [21]. Therefore, it seems reasonable to set up a separate place only for the patients treated for bone sarcomas, which may contribute to reducing the viral transmission and the number of infections.

Bone neoplasms are often a diagnostic challenge, and they can cause non-specific symptoms that can be easily confused with other common diseases [17,19]. Our study confirmed the data from the literature that the most common first symptom of bone tumor is deep pain (84%), which often increases over time and worsens at night. In addition, often, when the disease progresses, the patients report a palpable tumor or swelling (40%) and limb dysfunction (23%), and eventually, a pathological fracture (16%). There have been similar results reported George A. et al. where the pain was present in 88% of the patients with bone tumors, and there was a palpable tumor in 49% of the cases. However, they report fewer cases of a pathological fractures: 7% [17]. This difference in our study may result from a longer general delay in our group. Additionally, we observed a longer doctor-related delay by approximately 5.4 months in the group of patients with a pathological fracture (*p* = 0.037). This confirms the observations that before a pathological fracture occurs, the patients suffer an increasing level of pain and seek help from orthopedists or GPs. It is worth adding here that a pathological fracture in patients with bone neoplasm has an unfavorable prognostic factor, increasing the risk of distant metastasis, and reducing the overall survival [31,32]. Although bone sarcomas rarely cause general ailments such as fever, weakness or weight loss, in our study, approximately 6% of the patients reported these symptoms. Therefore, oncological vigilance and an in-depth diagnosis in the case of doubtful patients is crucial in detecting bone tumors. The diagnostic difficulty can also be seen in the example of radiological examinations, where in our study, only 44 to 63% of the X-rays gave a result suggesting a bone neoplasm. Although this examination is the gold standard in the diagnosis of bone tumors, especially in the initial stage of the disease, it is not a sensitive test, and it may give a false negative result [33,34]. This is also indirectly confirmed by our study, where X-rays detected bone neoplasms more often in the group where the tumor diameter was larger. The greatest difficulty in assessing the radiographs is seen in the cases of bone tumors that are located in the pelvis [19,34]. Although the diagnostic process of bone tumors is not an easy one, according to the literature, there are a number of actions that can be implemented to minimize the delay in diagnosis and treatment. Bone sarcomas are a rare type of cancer, which means that many physicians may not see these patients throughout their careers or encounter isolated cases [28]. Increased oncological vigilance among physicians, the inclusion of bone tumors in the differential diagnosis and knowledge of the most common symptoms may help to reduce the delay. In addition, in doubtful cases, the use of imaging tests that are more sensitive than classic radiographs are, such as MRI, CT or bone scintigraphy, can help detect bone neoplastic changes at the early stages. Moreover, the increase in the number of COVID-19 cases in the population should not be a reason for delaying the start of the diagnosis or treatment. As the data from the literature show, the patients with bone cancer can be safely treated even during a pandemic [20,21,22]. With use of appropriate measures, such as the isolation of infected patients, the use of personal protective equipment and vaccinations, the treatment of patients with bone neoplasms is safe both for the staff and the patients themselves [20,21].

Our study has certain limitations. First, the length of the delays in diagnoses during the COVID-19 pandemic may actually be much greater than we reported. As shown by some studies, Bone Sarcoma Treatment Centers observed a significant decrease in the detection of new cases [16,22]. Additionally, in our center, we observed a decrease in the number of oncological patients during the COVID-19 pandemic. Therefore, it can be suspected that some patients with bone tumors, due to the advanced disease process, were not eligible for surgery and were not referred to our center. It is therefore advisable to conduct a multicenter study that also includes patients who have been disqualified from surgery. In our study, however, we observed a trend that indicates a prolongation of the delay in diagnosis during the COVID-19 pandemic. The second important limitation is the retrospective nature of the study. In some of the patients, due to there being missing essential data in the medical records (including the symptoms and Patients Related delay), we obtained the information during an interview during a follow-up visit. Unfortunately, some of the patients, especially those treated before the pandemic, did not remember all of the information. However, in order to minimize this effect, the information obtained from the patient was compared to their medical records prior to admission at our center. The third limitation of our work is the lack of validation of the survey conducted among the patients treated during the COVID-19 pandemic. However, the entire questionnaire is based on previous studies, and it is designed to supplement the medical history interview with additional information about the course of the disease and the reasons for the delay in diagnosis [20,22].

## 5. Conclusions

Given the unprecedented impact of the COVID-19 on the healthcare systems around the world, a negative trend in the delay of diagnosis and treatment of bone neoplasms seems inevitable. We observed a prolongation of the pre-hospital and hospital-related diagnosis process in patients with bone tumors. This applies, in particular, to there being limited access to radiological examinations and a delay in performing biopsy. The patients treated during the pandemic presented significantly greater diameters of the bone lesions, which indirectly proves the negative impact of the COVID-19 pandemic on the diagnosis delay. Moreover, with appropriate measures, the patients with bone neoplasms can be safely diagnosed and treated during the COVID-19 pandemic.

## Figures and Tables

**Figure 1 cancers-14-06037-f001:**
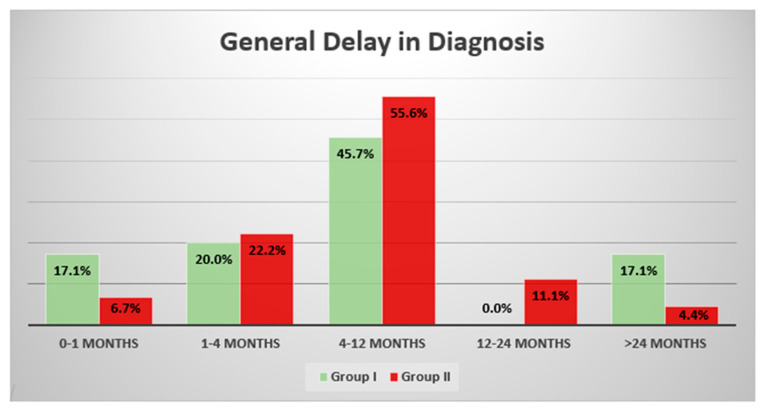
Graph presents the general delay in diagnosis of bone neoplasms. There were no significant differences between the groups (*p* > 0.05). Moreover, within the recommended delay time, which should not exceed 4 months, in Group I, this result was achieved by 37.1% of patients, and in Group II, it was achieved by 28.9% of patients (*p* = 0.821).

**Figure 2 cancers-14-06037-f002:**
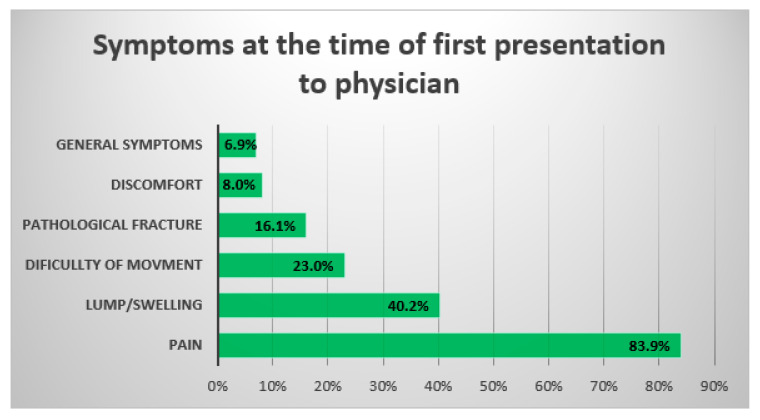
Symptoms reported by patients in both groups at the time of the first visit to the physician.

**Figure 3 cancers-14-06037-f003:**
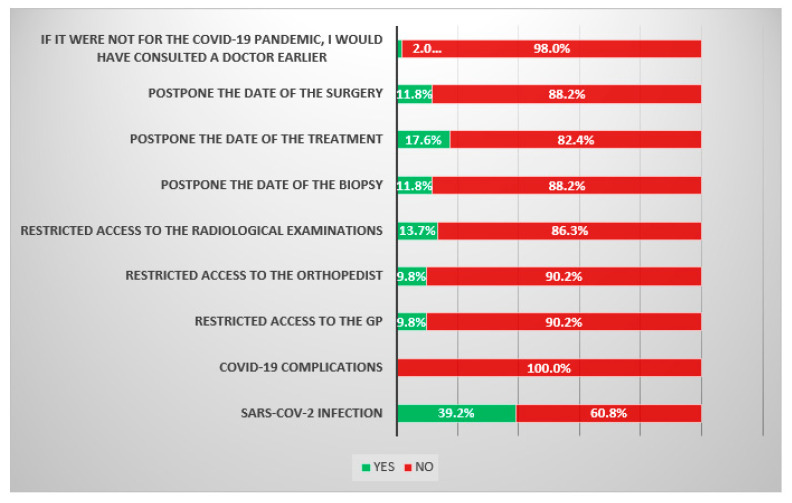
Survey results among patients treated during the COVID-19 pandemic. The survey consisted of 9 closed questions—Yes/No.

**Table 1 cancers-14-06037-t001:** Characteristics of patients treated for bone neoplasms before the pandemic (Group I) and during the COVID-19 pandemic (Group II). SD—Standard deviation; IQR—interquartile range.

	Group I (*n* = 36)	Group II (*n* = 51)	*p* Value
Gender - Male - Female	17 (47.2%) 19 (52.8%)	27 (52.9%) 24 (47.1%)	0.599
Age, mean (SD) (years)	40.7 (16.8)	45.6 (18.7)	0.215
Type of bone neoplasm - Osteosarcoma - Chondrosarcoma - Giant Cell Tumor - Other malignant neoplasms - Other benign neoplasm	8 (22.2%) 11 (30.6%) 6 (16.7%) 4 (11.1%) 7 (19.4%)	13 (25.5%) 16 (31.4%) 6 (11.8%) 12 (23.5%) 4 (7.8%)	0.923 0.878 0.736 0.116 0.102
Localization of the neoplasm - Lower Limb - Upper Limb - Pelvis and/or Sacrum	19 (52.8%) 9 (25.0%) 8 (22.2%)	25 (49.0%) 13 (25.5%) 13 (25.5%)	0.730 0.843 0.923
Diameter of bone lesion median (IQR) (mm) - Malignant lesions - Benign lesions	56 (38.8–70) 67 (54–79) 45 (34–50)	75 (47.5–115) 78 (50–120) 60 (41.3–101.3)	0.025 0.237 0.077

**Table 2 cancers-14-06037-t002:** Delay in diagnosis and results of further diagnostic and therapeutic pathway of bone neoplasms. The Mann–Whitney U test was used to compare the variables. IQR—interquartile range.

	Group I (*n* = 36) Median (IQR)	Group II (*n* = 51) Median (IQR)	*p* Value
- General delay in diagnosis (months)Malignant lesions - Benign lesions	7 (3.5–12) 6.5 (3–12) 7.5 (4–11.3)	10 (4–12) 10 (4–12) 10 (5.5–11.8)	0.728 0.861 0.692
Patient-related delay (months) - Malignant lesions - Benign lesions	1 (1–3) 1 (1–3.5) 1 (1–2.5)	2 (1–4) 2 (1–4) 1 (0.3–3.8)	0.619 0.556 0.884
Doctor-related delay (months) - Malignant lesions - Benign lesions	5 (2–10) 5 (1.5–9.5) 6 (3.5–16.5)	3 (1.5–8) 2.5 (1–7.5) 5 (2.5–11)	0.265 0.544 0.845
Time to perform biopsy (days) - Malignant lesions - Benign lesions	6.5 (2–10.3) 5 (2–8) 7 (5–12)	12 (2–27) 7 (2–26.5) 18 (13–28)	0.025 0.322 0.063
Time to histopathological result (days) - Malignant lesions - Benign lesions	12 (8.5–18) 15 (8–19.5) 10.5 (9.8–14)	13.5 (9–22.8) 13 (9–21.3) 20 (7–27.3)	0.450 0.750 0.440
Time to perform surgery (days) - Malignant lesions - Benign lesions	95 (38.8–70) 97 (78.5–102.3) 93 (73.5–94)	104 (48–148.5) 104 (86–147.5) 92.5 (44.8–144.3)	0.494 0.460 0.859

## Data Availability

Not applicable.

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
