# Peer review of "Delay in Diagnosis and Treatment of Primary Bone Tumors during COVID-19 Pandemic in Poland"

_cancers, 2022, doi:10.3390/cancers14246037_

Round 1

Reviewer 1 Report (Previous Reviewer 3)

Authors wanted to check the weak points of the diagnostic and therapeutic process of patients with bone tumors. We have shown that the COVID-19 pandemic had impact on diagnostic process of patients . In addition, authors shown that the lack of oncological vigilance among physicians and the insufficient sensitivity of classic radiographs still remain a weakness in this process. Therefore, I believe that this work indicates trends that may be helpful in creating further consensus in the field of bone cancer diagnosis.

Reviewer 2 Report (Previous Reviewer 2)

I am satisfied that the authors have addressed all of my previous concerns about the article. It is now much improved and I feel that it is now suitable for publication.

Reviewer 3 Report (Previous Reviewer 1)

my previous remarks were sufficiently addressed in the revised manuscript.

This manuscript is a resubmission of an earlier submission. The following is a list of the peer review reports and author responses from that submission.

Round 1

Reviewer 1 Report

General remarks: Kotrych et al. investigated diagnostic delay in primary bone tumors during CIVID-19 pandemic in Poland. The survey included adolescents form 16 yrs of age and adult patients. The impact of COVID-19 pandemics on oncology care has been studied and reported by several groups. Observations showed considerable variations between regions, countries, age groups and diagnostic groups. It seems that pediatric patients with cancer are less severely affected due to the pandemics than adults (see e.g. results from the same region, Central Europe: Müller J et al. Pathol Oncol Res. 2022 Mar 31;28:1610261.). The strength of the submitted manuscript is that authors investigated a relatively homogeneous group of patients with primary bone tumors. As they pointed out, diagnostic delay in these tumors, unfortunately, was quite frequent before COVID-19 pandemics, therefore, these patients represent a vulnerable group. Although authors, fortunately, could not demonstrate significant diagnostic delay in several comparisons, yet the trend was clearly present. Yet, in two important domains, i.e. in biopsy delay, and, quite interestingly, in “Doctor’s Delay” among patients with pathological fractures as compared to those patients without confirmed fractures. The poor impact of diagnostic delay was highlighted with the fact that tumor diameter was significantly greater in patients diagnosed in the COVID-19 pandemics period than in those before the outbreak of the pandemics. Although authors did not evaluate treatment outcome measures in detail, they noted that there were more amputation procedures performed among patients diagnosed during the pandemics than before.

Detailed minor remarks:

Line 26: the word “reported” was unnecessarily repeated.

Line 40: Adjust plural subject and singular predicate: “neoplasms was”

Lines 42-43: Last part of the last sentence before “Conclusion”: predicate is missing

Table 1, in row: “Diameter of bone lesion” and two rows below in row “Benign lesions” median values in Group II are not well separated from the ranges in Group I, edit!

Author Response

We would like to thank the reviewer for the comments on our manuscript. They have been very helpful and we hope that the manuscript will be more interesting to readers after the corrections. Below is our response to each point raised by the reviewer.

The text of the manuscript has been re-checked for grammatical correctness. In addition, we have corrected all the mistakes listed by Reviewer 1.

Reviewer 2 Report

Authors in this manuscript have aimed to determine how the pandemic affected the diagnostic and therapeutic process of patients with primary bone tumors. They also investigated whether surgical treatment could be safe for patients with bone tumors during the COVID-19 pandemic. Although study design and presentation of the results are quite appropriate for the scope of the manuscript, the quality of the manuscript is not sufficient for publication. I would like to offer the following points for consideration by the authors towards the improvement of the manuscript:

1- I think it would be better if the title includes not only the diagnosis but also the therapeutic process.

2- Introduction is very short and did not make enough argument about the necessity of the study. It is not clear to me whether similar studies have been done in the past, not necessarily limited to Poland or bone tumors.

3- They did not mention how the data collection tool and questionnaire were developed and if they test its validity.

4- In Table 1, explain why you specify median-range when referring to age and diameter (for example, why not mean-SD or median-IQR). Were these variables parametrically or nonparametrically distributed?

5- Monthly differences in new bone cancer diagnoses between 2018 and 2022 can be given as figure.

6- Since one of the goals of this study is to determine how we can prevent delay in diagnosis and treatment in the future, this should be mentioned in more detail in the discussion.

7-There are a number of grammatical/spelling errors which need correcting. Some examples include:

-P1 Line 26:  Please correct “those patients reported reported to oncology..”

-P1 Line 40: Please correct “ after X-ray examination, bone neoplasms was 40 suspected more frequently…”

-P3 Line 110: Please correct “at the time of the Pandemic announcement…”

8-Finally, some recent and relevant studies could enrich your manuscript.

- https://doi.org/10.1136/bmjspcare-2021-003301

-  https://doi.org/10.3390/ijerph18179098

Author Response

We would like to thank the reviewer for the comments on our manuscript. They have been very helpful and we hope that the manuscript will be more interesting to readers after the corrections. Below is our response to each point raised by the reviewer.

1- I think it would be better if the title includes not only the diagnosis but also the therapeutic process.

Thank you for this comment. The title has been changed:

"Delay in diagnosis and treatment of primary bone tumors during COVID-19 pandemic in Poland"

2- Introduction is very short and did not make enough argument about the necessity of the study. It is not clear to me whether similar studies have been done in the past, not necessarily limited to Poland or bone tumors.

When writing the manuscript, we did not want to lengthen the introduction and we focused on the most important issues. So that more space and time can be devoted to the discussion. However, we thank you for your comment and we have added a paragraph relating to the diagnosis of bone tumors, and expanded the entire introduction by adding further cited studies.

3- They did not mention how the data collection tool and questionnaire were developed and if they test its validity.

The validity of the survey was not tested before the study. However, the entire questionnaire is based on previous studies and is designed to supplement the medical history interview with additional information about the course of the disease and the reasons for the delay in diagnosis. 

The information has been added in the text.

4- In Table 1, explain why you specify median-range when referring to age and diameter (for example, why not mean-SD or median-IQR). Were these variables parametrically or nonparametrically distributed?

Thank you very much for these comments. The variables in Tables were nonparametrically distributed. The mean - SD should not be used in this case. We corrected this mistake and added median-IQR to the text and tables.

5- Monthly differences in new bone cancer diagnoses between 2018 and 2022 can be given as figure.

We added information in the text in the result section - 3.1. Participant characteristics.

6- Since one of the goals of this study is to determine how we can prevent delay in diagnosis and treatment in the future, this should be mentioned in more detail in the discussion.

In each paragraph  we discussed the weak points of bone neoplams diagnosis. However, we agree that it is worth discussing this issue in more detail. The discussion has been changed.

7-There are a number of grammatical/spelling errors which need correcting. 

The text has been checked again for grammatical correctness.

Corrections have been made

8-Finally, some recent and relevant studies could enrich your manuscript.

Thank you for recommending interesting articles. We have included them in our manuscript.

Reviewer 3 Report

This study does not appear to be substantial, and it is clear that COVID-19 will affect the management of other diseases.

Author Response

We would like to thank the reviewer for the comments on our manuscript. They have been very helpful and we hope that the manuscript will be more interesting to readers after the corrections. Below is our response to each point raised by the reviewer.

We fully agree with Reviewer 3 that it is clear that COVID-19 will affect the management of other diseases. In many centers of orthopedic oncology, patients report with an advanced form of bone sarcoma. On the one hand, our study shows what is visible in the practice of many oncologists and orthopedists. On the other hand, the one of aim of our study was to check what specific factors influenced the delay in diagnosis and treatment, apart from the COVID-19 pandemic itself. In addition, we wanted to check the weak points of the diagnostic and therapeutic process of patients with bone tumors. We have shown that the COVID-19 pandemic had impact on diagnostic process of patients . In addition, we shown that the lack of oncological vigilance among physicians and the insufficient sensitivity of classic radiographs still remain a weakness in this process. Therefore, we believe that our work indicates trends that may be helpful in creating further consensus in the field of bone cancer diagnosis.

The methodology, results and conclusions were checked again and corrections were made